# The Application of Clustering on Principal Components for Nutritional Epidemiology: A Workflow to Derive Dietary Patterns

**DOI:** 10.3390/nu15010195

**Published:** 2022-12-30

**Authors:** Andrea Maugeri, Martina Barchitta, Giuliana Favara, Claudia La Mastra, Maria Clara La Rosa, Roberta Magnano San Lio, Antonella Agodi

**Affiliations:** Department of Medical and Surgical Sciences and Advanced Technologies “GF Ingrassia”, University of Catania, 95123 Catania, Italy

**Keywords:** diet, dietary factors, dietary dataset, nutritional epidemiology

## Abstract

In the last decades, different multivariate techniques have been applied to multidimensional dietary datasets to identify meaningful patterns reflecting the dietary habits of populations. Among them, principal component analysis (PCA) and cluster analysis represent the two most used techniques, either applied separately or in parallel. Here, we propose a workflow to combine PCA, hierarchical clustering, and a K-means algorithm in a novel approach for dietary pattern derivation. Since the workflow presents certain subjective decisions that might affect the final clustering solution, we also provide some alternatives in relation to different dietary data used. For example, we used the dietary data of 855 women from Catania, Italy. Our approach—defined as clustering on principal components—could be useful to leverage the strengths of each method and to obtain a better cluster solution. In fact, it seemed to disentangle dietary data better than simple clustering algorithms. However, before choosing between the alternatives proposed, it is suggested to consider the nature of dietary data and the main questions raised by the research.

## 1. Introduction

One of the most important achievements of nutritional research in the last decades was the design of several large studies, including observational cohorts and clinical trials [1]. However, their results were often controversial, particularly for specific questions about the effect of vitamins and other micronutrients on cardiovascular diseases and cancers [2,3]. Some researchers interpreted these controversies as evidence of one of the main weaknesses of the single-nutrient approach against chronic diseases [1]. Indeed, clinical trials often evaluated the short-term effect of vitamin supplements in high-risk patients, while observational cohorts usually investigated the habitual intake of vitamins from diets in the general population. This, at least in part, revealed that single-nutrient approaches were insufficient to clarify many aspects of the effect of diet on chronic diseases [1]. For this reason, several researchers began to focus on the recognition of relevant dietary patterns characterized by higher intakes of fruits and vegetables, legumes, nuts, and whole grains rather than high-calorie and processed foods rich in sugar, salt, and additives [4]. The recognition of the importance of overall diet has initially driven forward research on popular and empirical dietary patterns (e.g., Mediterranean, vegetarian, vegan, anti-inflammatory, etc.) [4,5,6,7,8,9,10,11,12,13]. Only more recently, however, modern approaches to data analysis have become useful for the identification of a posteriori dietary patterns from available dietary datasets. Different dietary patterns have so far been derived using multivariate data analysis techniques, which are, in fact, specifically designed to identify meaningful patterns in complex multidimensional datasets [14,15,16,17,18,19,20,21,22,23,24,25,26,27,28]. Among these, principal component analysis (PCA) and cluster analysis represent the two most used techniques in nutritional epidemiology. In general, PCA works on a multidimensional dataset of correlated variables to reduce it into a low number of uncorrelated principal components (PCs) [29]. Working on a dietary dataset, these PCs can be used to derive different dietary patterns of the study population [16,17,18,19,30]. However, PCA does not create mutually exclusive patterns; rather, each individual receives a score of adherence to each dietary pattern derived. By contrast, cluster analysis assigns individuals to discrete groups, each featuring a reasonably coherent dietary pattern [31]. In nutritional epidemiology, both hierarchical clustering and K-means clustering have found wide applications so far [32,33]. Moreover, there are also a lot of studies that have applied these techniques in parallel, with the aim of discovering differences between dietary patterns obtained [34,35,36,37,38].

In the current work, we propose a workflow to combine PCA, hierarchical clustering, and K-means algorithm in a novel approach for dietary pattern analysis. This approach, defined as the clustering of PCs, could be useful to leverage the strengths of each method and to obtain a better cluster solution [39]. Here, we explain the meaning of each step in the workflow, also providing some alternatives in relation to different dietary data used. Moreover, we also give an example of how to apply this approach to an existing dietary dataset and compare cluster solutions with those obtained by simple hierarchical clustering and K-means clustering.

## 2. Materials and Methods

### 2.1. Study Design

Figure 1 summarizes the crucial steps of the workflow, from data collection to clustering. As an example, a clustering of PCs was applied to a dietary dataset obtained by integrating information from women referring to two clinical laboratories or to the cervical cancer screening unit of Catania (Italy) for routine examinations. Women were selected from those participating in three epidemiological studies from 2010 to 2017. Although the studies were carried out in different periods, they shared similar objectives and the same protocols and methods. For the purpose of the current analysis, it is important to note that the tool used for the dietary assessment was the same between studies and that additional information on protocols and methods was fully reported elsewhere [30,40,41,42,43]. The current analysis was performed on data from 855 non-pregnant women aged 15–85 years without a history of severe diseases (i.e., cancer, cardiovascular diseases, diabetes, neurodegenerative, and autoimmune diseases). All women were informed about the study and signed an informed consent statement. All the studies were conducted in accordance with the Declaration of Helsinki, and the study protocols were approved by the ethics committees of the involved institutions.

### 2.2. Data Collection

The first question to ask when approaching nutritional research is what kind of dietary data are needed to achieve the objective. Indeed, it is not always necessary to collect new data since a lot of dietary datasets are made openly available in research archives and data repositories. Although using existing datasets can save a lot of time and effort, this is not always possible, and it is necessary to collect data through ad-hoc tools. It should be recalled that all dietary assessment tools are—to some extent—imperfect, with each suffering from some limitations [44]. For instance, dietary recalls in general and, in particular, 24-h recalls have the advantage of quantifying daily dietary intakes. However, they also have high investigator costs and assess dietary intakes in a very narrow window. By contrast, diet histories and food frequency questionnaires (FFQs) are retrospective tools that can capture habitual dietary habits with lower investigator costs, but they are more prone to inaccuracies and misreporting [44]. To overcome some of these limitations, alternative and complementary tools have been proposed in recent years; however, they are still in their infancy and not widely used [45]. With these considerations in mind, the understanding of the pros and cons of each dietary assessment tool, as well as a thorough exploration of the available data, are necessary to identify the optimal approach to analyze the dataset. For clarification purposes, it is important to underline that habitual dietary data should be used—rather than working on point measures—if the aim of the research is to derive dietary patterns.

In the current example, we use dietary data obtained through a 95-item semi-quantitative FFQ referred to the month preceding the recruitment. The structure of this FFQ has already been described elsewhere [19,42,43,46]. In brief, participants were asked to report the monthly frequency of consumption and portion size related to 95 foods with the support of a photographic atlas. Frequencies of consumption and portion sizes were used to calculate daily intakes of foods, expressed as grams per day. The 95 foods were grouped into 39 predefined food categories, reflecting similarities in nutrient profile and/or culinary use. Individual food items constituting a distinct item or characterizing a particular dietary pattern were preserved.

### 2.3. Data Cleaning

Data cleaning is a fundamental element of data analysis and consists of the process of identifying missing or inaccurate data and then handling each issue by imputation or remotion. The next steps of our workflow will require a dataset without missing data and potential outliers that, if present, can affect the PCA and the subsequent clustering. As a general rule of thumb, records with more than 5% missing data should be removed. Otherwise, the blanks can be filled with median or median values of non-missing data or alternatively by applying dedicated algorithms. For example, the K Nearest Neighbors (KNN) algorithm is widely used to replace missing data in similar contexts [47]. Beyond excluding impossible data and entry errors, dealing with outliers represents a critical step in our workflow as it presents some alternatives. Indeed, there are so many methods for outlier detection that it is not possible to suggest one of them. The choice depends on the data type, the accuracy of the dietary assessment tool used, and the suspected quantity of outliers. Moreover, some methods can be more time-consuming than others.

In our example, we propose to calculate the daily energy intake for each participant and then apply Tukey’s method for detecting outliers. This method is based on the graphical depiction of data through the box plot. Each value of daily energy intake that was above or below the whiskers was removed. It is worth remembering that whiskers correspond to 1.5 times the interquartile range (IQR) subtracted from the 1st quartile or added to the 3rd quartile. Only after having removed implausible daily energy intakes, we inspect potential outliers in other food categories.

### 2.4. Data Transformation

Working on a dietary dataset, it might not be necessary to standardize data since they are often measured on the same scale. However, if there are large differences between the ranges of continuous variables (e.g., a food category ranges from 0 to 100 g per day while other ranges from 0 to 1), some preprocessing with normalization or standardization could be useful. These can be respectively done by scaling the variables to a desired range or by subtracting the mean and dividing by the standard deviation for each value of each variable (i.e., z-score). Moreover, it is necessary to check the assumption that the relationships between variables are linear. If this assumption is violated, data log-transformation or its alternatives are strongly recommended. Finally, it is important to note that adjustment for kilocalories could be useful for analyzing data from participants with different daily energy intakes. Ideally, this should be done prior to data transformation with one of the following approaches: the density method rescales dietary data as a proportion of total energy intake; the residual method indirectly adjusts for kilocalories by regressing dietary intakes on the total energy intake [48,49]. An alternative is to control for the effect of total energy intake a posteriori in a multivariable risk model with the outcome of interest. However, dietary patterns obtained would remain affected by differences in total energy intake.

In our example, dietary intakes were adjusted for daily energy intake using the residual method [48], and then energy-adjusted data were standardized to their z-scores.

### 2.5. Principal Component Analysis

PCA has so far been applied in nutritional epidemiology to simplify the complexity of high-dimensional datasets obtained through FFQ, dietary records, or dietary history questionnaires [50]. Specifically, PCA is commonly used to reduce a dietary dataset of a set of correlated variables into fewer dimensions reflecting distinct dietary patterns. These dimensions–generally called PCs—are rank-ordered by total variance explained, uncorrelated, and fewer in number than the initial variables [29]. Prior to applying PCA to a dataset, some assumptions should be verified. Firstly, the sample size should be large enough to produce a reliable result. Secondly, there should be adequate correlations between variables to be reduced to a few numbers of PCs. The method used to detect sampling adequacy is the Kaiser-Meyer-Olkin (KMO) Measure of Sampling Adequacy, while Bartlett’s test of sphericity is applied to test the hypothesis that the correlation matrix is an identity matrix. In general, high values of the KMO (i.e., close to 1) and small values of Bartlett’s test (less than 0.05) indicate that the dataset is suitable for PCA.

Without discussing the mathematics underlying PCA, it derives for each PC the eigenvalue and the eigenvector from the covariance matrix, which respectively represent the total amount of variance explained and its orientation [29]. In the case of untransformed data with different scales, the alternative is to use the correlation matrix as the input to PCA [29]. The number of PCs to be retained for further analyses is usually determined according to eigenvalues and the amount of variance explained [18,30]. In general, there are several rules of thumb to determine an acceptable number of PCs (e.g., those reaching a cumulative variance of ~ 80% or those with eigenvalues >1). However, most of these criteria do not apply well to nutritional epidemiology. For example, the percentage of variance explained is usually between 10% and 30%, while the cut-off value for eigenvalues is around 1.6 [32,35]. This is to retain a few numbers of PCs to be interpreted and analyzed. A third criterion is based on the visual inspection of the Scree plot (i.e., a line plot of the eigenvalues of PCs) for identifying an elbow, following which subsequent PCs add little to the variance explained. Each PC can be interpreted in terms of correlations with initial variables, which are represented by the PC loadings. To simplify the interpretation of PCs, the varimax rotation is usually applied, and individual PC scores are generated [16,17,19].

In the current example, we applied the PCA with varimax rotation on the covariance matrix of standardized and energy-adjusted dietary data. In the main analysis, we retained the first two PCs according to the Scree plot examination. However, we also evaluated changes associated with retaining the first four PCs or all PCs with eigenvalues >1.

### 2.6. Clustering and Consolidation

The next step of our workflow is to apply hierarchical clustering to selected PCs in order to reveal different clusters based on the hierarchical tree. In this agglomerative clustering, the data points are iteratively merged based on their pairwise distance [31]. Although there are many methods to measure this distance, we suggest using Ward’s Linkage because it is based on the multidimensional variance in a way similar to PCA. There are several methods to choose the number of clusters to be generated. The first one evaluates the inertia (i.e., the mean squared distance between each instance and its closest centroid) with the increasing number of clusters. The second one decides the number of clusters that maximizes the silhouette score [31].

The clustering solution obtained by hierarchical clustering is further consolidated by K-means clustering. Since K-means clustering requires a predefined number of clusters, the algorithm is applied considering the number of clusters defined through hierarchical clustering. It is worth noting that the two methods could lead to slight differences in the clustering solution [31]. The agreement between hierarchical and K-means clustering can be assessed using the Adjusted Rand Index (ARI).

### 2.7. Statistical Analysis

All the steps above were performed using the SPSS (version 26.0, SPSS, Chicago, IL, USA), but it is also possible to use other software or existing libraries for common programming languages (e.g., Stata, R, Python). Next, we used the one-way analysis of variance (ANOVA) to compare z-scores of dietary intakes across clusters. The cluster solution obtained using hierarchical clustering of PCs was also compared to clusters obtained using simple hierarchical clustering or K-means clustering. Agreement between different cluster solutions was expressed as ARI. Unless otherwise indicated, all statistical tests were two-sided, and *p*-values < 0.05 were considered statistically significant.

## 3. Results

After data cleaning, the dataset used in the current example included 841 records without missing values or outliers for the 39 food categories under investigation. We first verified the assumptions of sampling adequacy (KMO = 0.758) and sphericity (*p*-value for Bartlett’s test < 0.001). Prior to analysis, dietary data were adjusted for total energy intake, standardized, and then subjected to PCA.

Thus, the dataset of 39 interrelated and correlated variables was reduced to 15 PCs with eigenvalue > 1, which cumulatively explained 58.8% of the total variance. In Appendix A, we depict the factor loadings for each PC, which reflect how initial variables loaded on PCs. The number of PCs to be retained was selected by inspection of the Scree plot and eigenvalues. However, the Scree plot revealed two elbows (Figure 2): the first elbow was after PC2, while the second one was after PC4. In the Appendix A, we also presented three Scree plots that illustrated how participants were distributed on the first four PCs (Appendix A). According to the Scree plot, the first option was to consider only PC1 and PC2, which cumulatively explained 15.5% of the total variance. However, as we will see later, we also considered retaining the first four PCs (which cumulatively explained 24.6% of the total variance and all PCs with eigenvalue > 1.

Thus, applying the hierarchical clustering to PC1 and PC2, we obtained the dendrogram shown in Appendix A, which facilitated the interpretation of clusters within the dietary dataset. Some information can be deduced from the dendrogram, such as the dissimilarities between clusters, which were represented by the branch size that linked them. To select the number of clusters to be retained, we calculated the Silhouette score for each cluster solution, and we opted for the one providing the highest value (Appendix A). In Figure 3a, we show the Scree plot of PC1 and PC2, in which participants were assigned to 3 different clusters. Although the plot already revealed a clear separation between clusters, we also consolidated the clustering solution by applying the K-means algorithm (Figure 3b). Specifically, most participants were assigned to the same cluster, while others were reassigned. The partial agreement between the two cluster solutions was also demonstrated by an ARI of 0.496.

Figure 4 shows the average z-scores for each food category and each cluster obtained through hierarchical clustering of PCs, simple hierarchical clustering, and K-means clustering. Notably, the hierarchical clustering of PCs led to three clusters with peculiar features: cluster 1 (*n* = 82) was characterized by a higher intake of boiled potatoes, vegetables, soup, legumes, and fish; cluster 3 (*n* = 183) was characterized by higher intake of red and processed meat, vegetable oil, sweets, dipping sauces, salty snacks, and fries; cluster 2 (*n* = 576) appeared to be a mixed and balanced group without a particular preference for specific foods. Instead, both simple hierarchical and K-means clustering led to a three-cluster solution where the first cluster shared some features with what was described above, while the remaining clusters were difficult to interpret. The low agreement of hierarchical clustering of PCs with simple hierarchical clustering and K-means clustering was confirmed by an ARI of 0.102 and 0.151, respectively.

Figure 5 shows the average z-scores for several nutrients across clusters obtained through the hierarchical clustering of PCs. In line with the findings presented above, cluster 1 was characterized by a higher intake of magnesium, folate, vitamin A, vitamin C, and vitamin D. Cluster 3, instead, was characterized by a higher total energy intake, as well as a higher intake of saturated and unsaturated fatty acids. Here too, cluster 2 seemed to be a mixed group with a slightly lower intake of minerals and vitamins. The average z-scores obtained by working on the first four PCs and on all PCs with an eigenvalue > 1 are reported in Appendix A. It is important to note that both options led to a two-cluster solution that differed from what was obtained previously. Indeed, in both cases, cluster 1 was mixed without any preference for specific foods, while cluster 2 was characterized by differing dietary features. This once more pointed out the crucial step of selecting the number of PCs to be retained.

## 4. Discussion

Our work provides evidence on applying clustering to PCs to derive dietary patterns from an existing dietary dataset. Although similar approaches have been previously applied in other fields of research [51,52], to our knowledge, our study is the first in nutritional epidemiology. The rationale behind our choice to combine three multivariate data analysis techniques was to leverage the strengths of each of them [39].

In general, the results of the two techniques are somewhat different in the sense that PCA helps to reduce the number of “features” while preserving the variance, whereas clustering reduces the number of “observations.” Thus, if the dataset consists of N observations and T features, PCA aims at compressing the T features, whereas clustering aims at compressing the N observations. Indeed, PCA is a non-supervised technique that helps us in searching for patterns in a multidimensional dataset and in selecting the minimum number of PCs accounting for the maximum variance [29]. However, findings from PCA are less easy to be interpreted than those obtained from cluster analysis. For instance, an intrinsic issue of PCA pertains to the fact that its output does not refer to a distinct group of individuals with different dietary habits but rather gives different scores of adherence to each dietary pattern characterizing the study population. Only later can individuals be classified in terms of adherence to each dietary pattern obtained [35]. Moreover, it may be difficult to disentangle different foods that characterize each PC and, thus, each dietary pattern. This was not, however, the case of PCs obtained in our example, in which the first two PCs were respectively characterized by plant-based foods and by typical products of the western diet. Cluster analysis is also a non-supervised technique, but it produces an output that can be easier interpreted [31,53,54]. In general, it allows the categorization of individuals in distinct clusters based on the degree of shared characteristics so that individuals in the same cluster are more similar than those included in other groups [55,56]. One benefit of hierarchical clustering is its intrinsic principle of segregating individuals on several hierarchical levels, which can be easily visualized in a dendrogram. On the other hand, however, the choice of the level at which cutting the dendrogram may produce clustering solutions with different meanings [31].

The novelty of our approach relies on the hypothesis of taking advantage of the characteristics of PCA and clustering to improve the analysis of dietary data. In particular, the reason for using a dimensionality reduction step—such as PCA—prior to data segmentation is to improve the performance of the clustering algorithm. The preliminary application of PCA, in fact, decreases the number of features to be analyzed and the noise. Accordingly, in our example, we first applied hierarchical clustering to PCs obtained by PCA of the original dataset. Among different clustering solutions, the best number of clusters to be retained was chosen according to their Silhouette scores. Next, the solution obtained using hierarchical clustering was consolidated with K-means clustering. This allowed us to balance the number of individuals in the two extreme clusters while maintaining their meaning. After doing so, we obtained three clusters that were clearly characterized by different dietary habits: the first one was assimilable to a healthy dietary pattern rich in fruits, vegetables, legumes, and fish; the second one appeared as a mixed and balanced dietary pattern; the third one, instead, represented a typical western diet rich in high-calorie and processed foods. These clusters, besides being similar to those obtained in previous studies [16,28,30,57], were clearly associated with different intakes of calories, fatty acids, vitamins, and minerals.

Thus, our approach seemed useful to distinguish individuals based on their intake of foods and nutrients. In particular, it produced separate clusters and improved the interpretation of findings if compared to PCA alone, which instead does not separate observations into clusters. As demonstrated by our findings, the combined approach also produced a better solution than those obtained by simple hierarchical clustering or K-means clustering. In fact, the cluster featuring high-calorie and processed foods emerged only from the clustering of PCs but not from simple clustering. For these reasons, our approach appeared promising for further downstream applications, such as studying the main determinants of dietary choices or investigating the association between diet and chronic diseases. Despite these strengths, it would be appropriate to consider weaknesses that characterize the wide range of techniques currently used to derive dietary patterns in a study population. As for all data-driven methods, the outputs, and hence any difference between different techniques, may depend on the dietary dataset. Moreover, a full understanding of dietary data to interpret is important in the various options to choose through the entire process of analysis. Accordingly, our suggestion is to decide the better technique to be used on a case-by-case basis.

To guide the user in applying our approach, we have provided some of the possible alternatives for all the steps of data management, from data cleaning to multivariate analysis. The choice between different options, but also the application of other common tools not described above, should be assessed on a case-by-case basis, depending on the nature of the data and on research questions. In our example, we only compared clustering solutions obtained by selecting a different number of PCs prior to clustering. The solution based on the first two PCs differed significantly from those obtained from working on four PCs or on all PCs with an eigenvalue > 1. However, it was also the best in terms of Silhouette score and cluster interpretability. Thus, our suggestion is to select the number of PCs to be retained only after inspection of the Scree plot and, if possible, after comparing solutions from different selections.

## 5. Conclusions

In summary, the present study describes an alternative pipeline to derive dietary patterns, which combines three of the most commonly used multivariate data analysis techniques in nutritional epidemiology. Our integrative approach seems to disentangle dietary data better than simple clustering algorithms, discovering dietary patterns that reflect those that have been generally obtained by previous studies. Although our workflow overcomes some limitations of other techniques, it also presents certain subjective decisions that might affect the final clustering solution. Thus, as stated above, we suggest considering the nature of dietary data and the main questions raised by the research before choosing between different techniques and alternatives to analyze dietary data.

## Figures and Tables

**Figure 1 nutrients-15-00195-f001:**
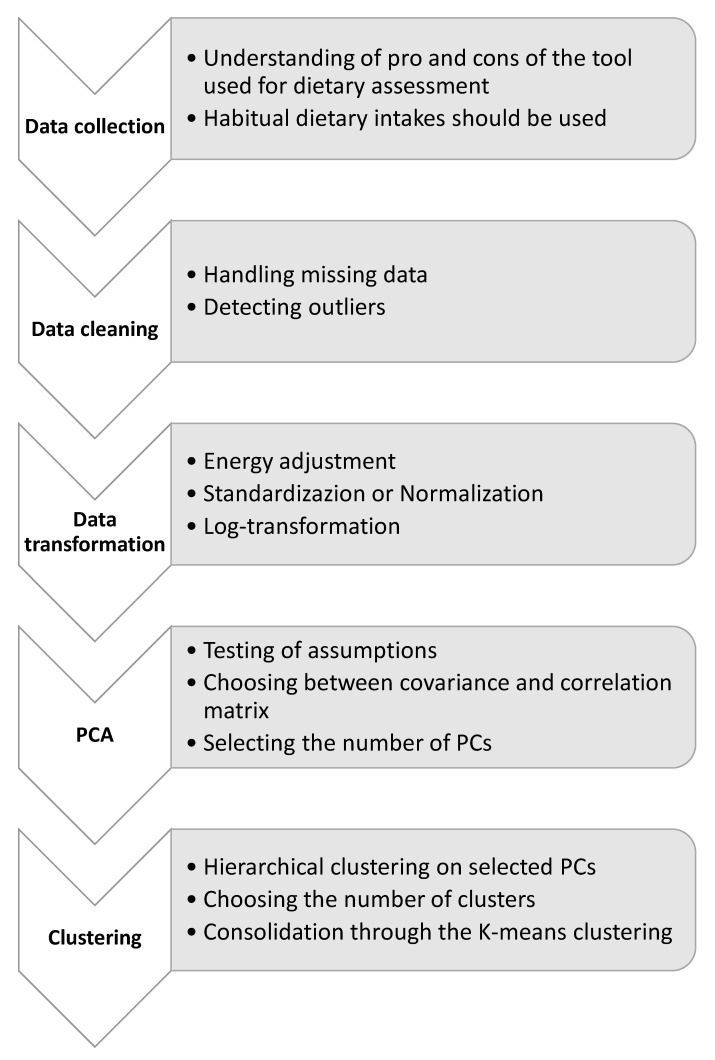
Workflow of the clustering on principal components to derive dietary patterns. The flow chart summarizes the steps of this approach, from data collection to clustering and consolidation. For each step, objectives and potential alternatives are reported. Abbreviations: PCA, principal component analysis; PCs, principal components.

**Figure 2 nutrients-15-00195-f002:**
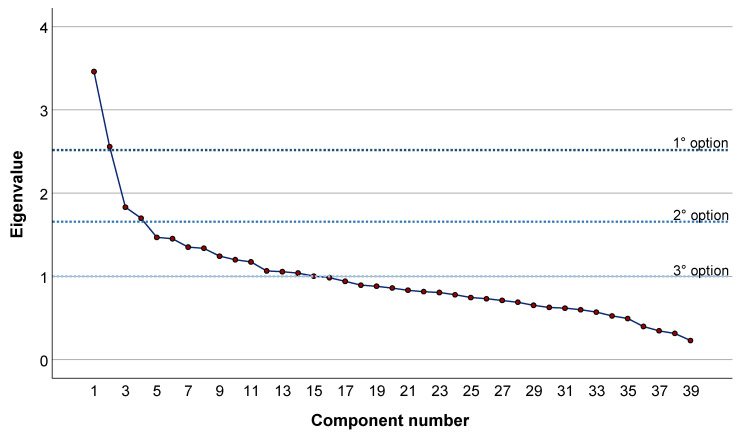
Scree plot of eigenvalues against the corresponding number of principal components. Dotted lines represent three options for selecting the number of principal components to be retained.

**Figure 3 nutrients-15-00195-f003:**
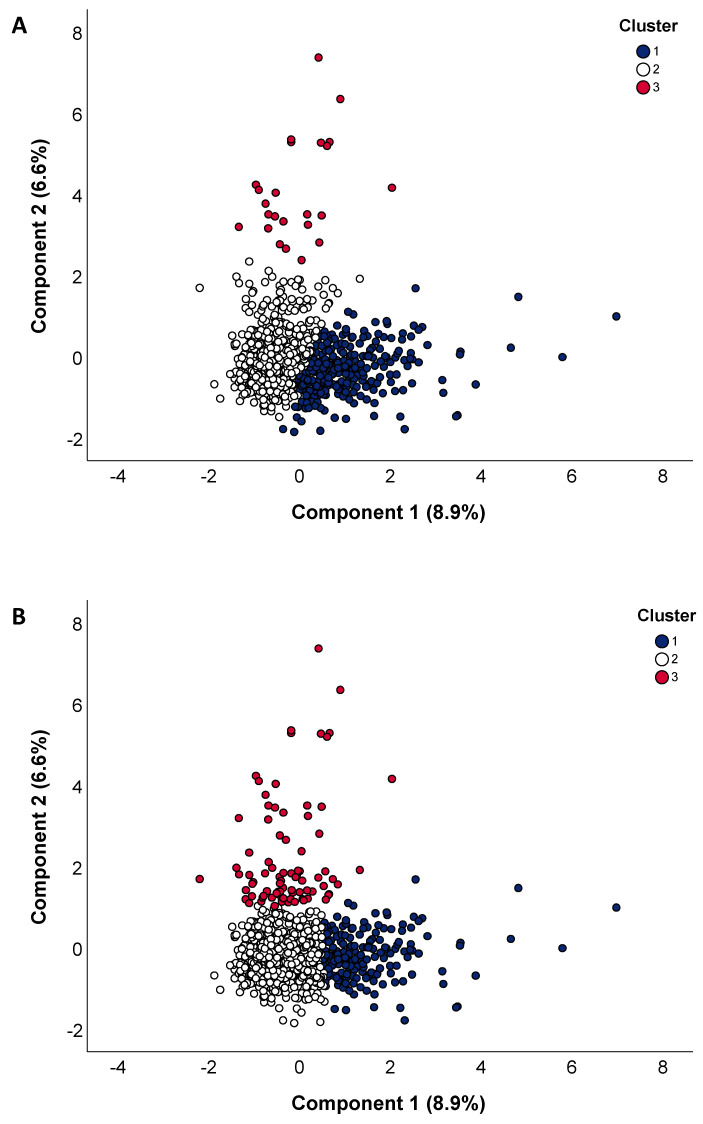
Outputs of the clustering of principal components. The score plots show the relationship between principal components 1 and 2, in which participants were assigned to 3 different clusters based on hierarchical clustering (**A**) and K-means clustering (**B**).

**Figure 4 nutrients-15-00195-f004:**
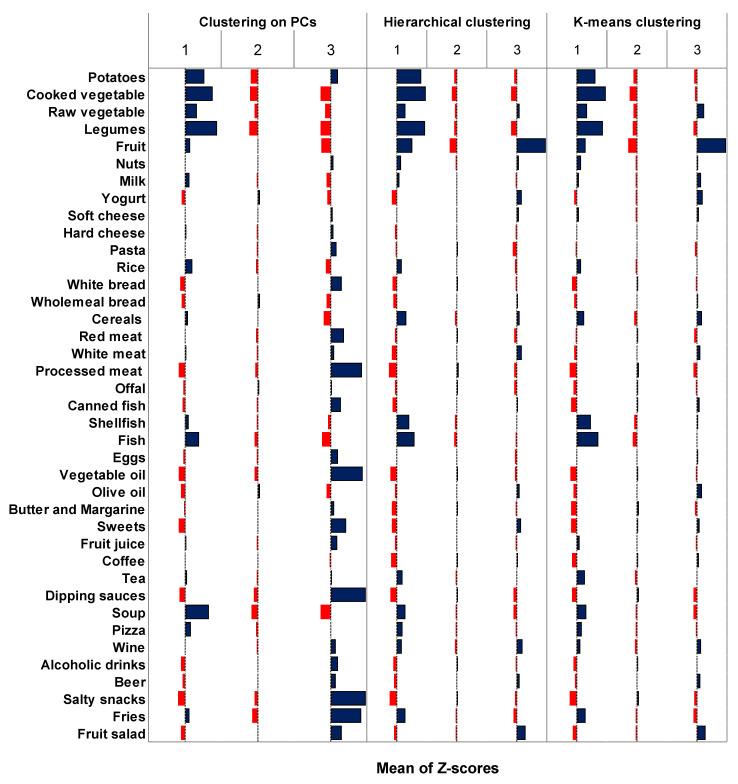
Comparison of z-scores of food intakes between clusters obtained through clustering of principal components, simple hierarchical clustering, and K-means clustering. Blue bars represent food categories that positively characterize the cluster. Red bars represent food categories that negatively characterize the cluster. Abbreviations: PCs, principal components.

**Figure 5 nutrients-15-00195-f005:**
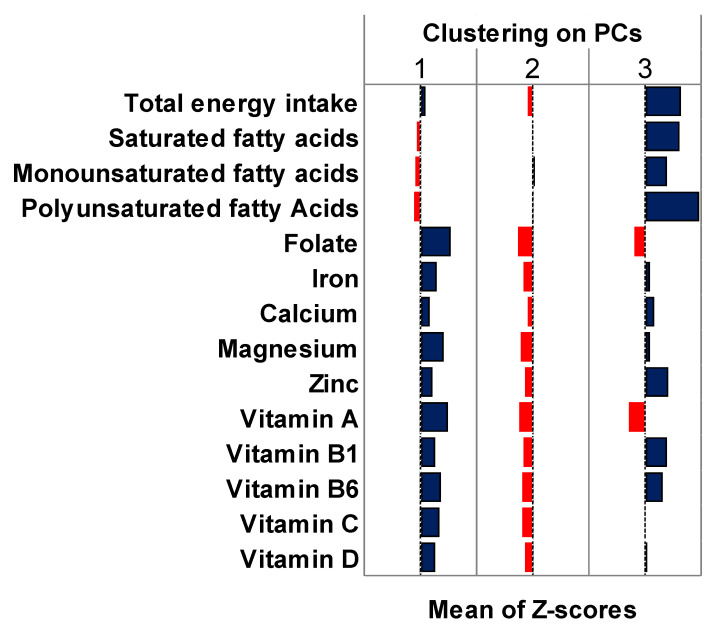
Comparison of z-scores of nutrient intakes between clusters obtained through clustering of principal components. Blue bars represent nutrient intakes that positively characterize the cluster. Red bars represent nutrient intakes that negatively characterize the cluster. Abbreviations: PCs, principal components.

## Data Availability

The datasets analyzed during the current study are available from the corresponding author upon reasonable request.

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
