# Peer review of "The Application of Clustering on Principal Components for Nutritional Epidemiology: A Workflow to Derive Dietary Patterns"

_nutrients, 2022, doi:10.3390/nu15010195_

Round 1

Reviewer 1 Report

Period of recruitment is to long so that is unbelievable that the same informed test was applied.

So many the same sentences repeated abouth methods of measurment and very poor information about selected persons.

Author Response

Dear Editor,

Please consider the revised version of the manuscript entitled “The Application of Clustering on Principal Components for Nutritional Epidemiology: a Workflow to Derive Dietary Patterns” in which we have considered all comments and suggestions from reviewers. This letter is intended for the convenience of the editor and reviewers and contains the list of the requested changes. The following list of changes and answers to comments of Reviewers addresses all revisions made in the manuscript (in red font).

Reviewer 1

R: Period of recruitment is to long so that is unbelievable that the same informed test was applied. So many the same sentences repeated about methods of measurement and very poor information about selected persons.

Answer: We would like to take this opportunity to thank the Reviewer for his/her comments and suggestions which helped us in improving our manuscript. We apologize if this point was not clearly described and please consider changes in the revised version of our manuscript. 

In our work, the combined approach of clustering on PCs was applied to a dietary dataset obtained by integrating information from women referring to two clinical laboratories or to the cervical cancer screening unit of Catania (Italy) for routine examinations. Women were selected from those participating in three epidemiological studies from 2010 to 2017. Despite the studies were carried out in different periods, they shared similar objectives and the same protocols and methods. For the purpose of the current analysis, it is important to note that the tool used for the dietary assessment was the same between studies and that additional information on protocols and methods were fully reported elsewhere. The current analysis was performed on data from 855 non-pregnant women aged 15–85 years, without history of severe diseases (i.e., cancer, cardiovascular diseases, diabetes, neurodegenerative and autoimmune diseases). All women were informed about the study and signed an informed consent. All the studies were conducted in accordance with the Declaration of Helsinki and the study protocols were approved by ethics committees of involved institutions.

Reviewer 2 Report

Thanks for presenting the novel method to process nutritional epidemiological data. But I do feel it needs more discussions. 

I totally agree that there is a possibility to use the combination of the two methods, however, you just tested one of the possibilities. How about using the PCA after the clustering? How about using one methods for multiple times? Even using PCA-clustering-PCA? Why do you choose the current sequence? 

Secondly, I am questioning the meaning of combination. There is no strong evidence to prove the improvement. I agreed that the combination of the two methods may bring the pros of both methods, but there is also a potential to get the cons of both. 

I appreciate the novel thinking of the authors, however, a further research and discussion would be necessary before publishing the work.

Author Response

Dear Editor,

Please consider the revised version of the manuscript entitled “The Application of Clustering on Principal Components for Nutritional Epidemiology: a Workflow to Derive Dietary Patterns” in which we have considered all comments and suggestions from reviewers. This letter is intended for the convenience of the editor and reviewers and contains the list of the requested changes. The following list of changes and answers to comments of Reviewers addresses all revisions made in the manuscript (in red font).

Reviewer 2

R: Thanks for presenting the novel method to process nutritional epidemiological data. But I do feel it needs more discussions.

Answer: We would like to take this opportunity to thank the Reviewer for his/her comments and suggestions which helped us in improving our manuscript.

R: I totally agree that there is a possibility to use the combination of the two methods, however, you just tested one of the possibilities. How about using the PCA after the clustering? How about using one method for multiple times? Even using PCA-clustering-PCA? Why do you choose the current sequence?

A: Dear Reviewer, thank you very much for this comment and apologize if this point was not so clear. In our work, we have combined PCA and clustering, as the two most commonly used techniques for the analysis of dietary data. While there are a lot of studies that have applied these techniques in parallel, to our knowledge no studies combined them in a single workflow.

In general, the results of the two techniques are somewhat different in the sense that PCA helps to reduce the number of "features" while preserving the variance, whereas clustering reduces the number of "observations". Thus, if the dataset consists of N observations and T features, PCA aims at compressing the T features whereas clustering aims at compressing the N observations.

In other fields, it is a common practice to apply PCA before a clustering algorithm, improving the clustering results in practice. By contrast, there are no reasonable evidence and reasons to use PCA after clustering or to apply one method for multiple times.

The novelty of our work relies on the hypothesis of taking advantage from the characteristics of these techniques to improve the analysis of dietary data. In particular, the reason for using a dimensionality reduction step - such as PCA - prior to data segmentation is to improve the performance of the clustering algorithm. The preliminary application of PCA, in fact, decreases the number of features to be analyzed and the noise.

Please consider integrations in the discussion section explaining the reason behind our choices.

R: Secondly, I am questioning the meaning of combination. There is no strong evidence to prove the improvement. I agreed that the combination of the two methods may bring the pros of both methods, but there is also a potential to get the cons of both. I appreciate the novel thinking of the authors, however, a further research and discussion would be necessary before publishing the work.

A: Dear Reviewer, as just commented in the previous point, the main reason behind our choice is to improve the clustering solution by reducing the number of features and the noise. Moreover, our combined approach produces separate clusters and improves the interpretation of findings if compared to PCA alone, which instead does not separate observations into clusters but rather gives different scores of adherence to each dietary pattern. In addition, as demonstrated by our findings, the combined approach produced a better solution than those obtained by simple hierarchical clustering or K-means clustering. In fact, the cluster featured by high-calorie and processed foods emerged only from clustering on PCs but not from simple clustering. As data-driven approaches, it is also true that

the outputs and hence any difference between different techniques may depend on the dietary dataset. For all these reasons, our suggestion is to decide the better technique to be used on a case-by-case basis.

Please consider integrations in the discussion section explaining the meaning of the combined approach.

Reviewer 3 Report

Esteemed Authors,

 It has been a great honor, as well as a pleasantly challenging activity, to review the article entitled The Application of Clustering on Principal Components for Nutritional Epidemiology: a Workflow to Derive Dietary Patterns.”

Agricultural and food production has significantly changed in the last 20 years. Obviously and logically, increasing population and consumption put pressure on the world's food supply. Accordingly to the data of the Food and Agriculture Organisation (FAO), by the effect of demographic growth and changes in diets and incomes, the demand for food will likely grow by 70% until 2050. The current outlook of the increasingly global market is marked by considerable uncertainties of supply linked to unpredictable economic, political, climatic, and biological developments. This implies a need for accelerated agricultural production growth in developing countries.

The list of new challenges is open and includes the most unexpected situations, from new crop and animal diseases to significant climate change and emerging diseases.

Current agriculture consumes enormous resources for development: over 70% of freshwater reserves are used for agriculture. At the same time, agriculture represents an essential threat to the environment: it is responsible for polluting almost 80% of the oceans and freshwater reserves. On the other hand, products of animal origin are among the top products in terms of greenhouse gas emissions, and approximately one-third of the food produced in the world for human consumption — nearly 1.3 billion tons — gets lost or wasted every year.

With a few exceptions, food production is growing. However, the challenges to primary production are increasing. Concerning organic farming in the European Union, the demand for such products is constantly growing, and the implementation mechanisms need to be continuously adjusted.

The recent reforms of CAP (Common Agricultural Policy) and other EU policies and international and bilateral trade negotiations take into account the objective of global food security. The  Joint Research Center (JRC) of the European Commission is involved in the impact assessment of policies regarding food security. Also, the potential trade agreements through economic modeling and the global CGE (Computable General Equilibrium) models assess the economy-wide impacts of the trade policy changes. All these changes are affecting all sectors of the partners. Besides, the global partial equilibrium models simulate the consequences incurred by the agricultural areas of the partners.

The situation is much more complicated concerning organic products than conventional products. Given that organic products are among the top products subject to various frauds, consumer confidence appears as an essential element of the development of the food chain.

Theory-wise, the paper will likely elicit specialists' interest in consumer behavior, sociology, sustainable development of agriculture, nutrition, food processing, food consumption, public policies, and public health.

The paper is well structured and possesses an appreciable novelty character. The main components of the article – Introduction, Materials and Methods, Results, Discussion, and Conclusions - are organized judiciously and directly linked to one another.

The documentation is adequate, and the provided scientific results are precise. The goal of the conducted research is well-specified and delineated. The working protocol is appropriate, and the analysis methods are coherent with the proposed objectives.

The bibliography of the paper is generous. What is even more relevant for the overall quality of the article, despite the appreciable number of bibliographic sources, all the authors in the bibliographic reference list are quoted in the text of the material (without exception).

The article is very well documented, and most bibliographic references are recent and very recent.

I would advise the authors to be more careful concerning the bibliography: it is preferred to mention the authors in alphabetical order, and references without specified authors are cited at the end of the list of references in chronological order. I also recommend using a single system not only in citations but also when it comes to journals. I am referring here mainly to mentioning the following elements for each article consulted: journal, volume, issue, and pages. Supplementary, the DOI may also be noted if the authors desire, but the essential descriptive elements are the previously mentioned ones.

Also, to avoid confusion, it is recommended to accurately mention the article's descriptive elements - for example - the additional mention of the article number, where the situation requires it.

For example – page 14, line 506, number 47 in the bibliographic references list – Malarvizhi, R., Thanamani, A.S. K-Nearest Neighbor in Missing Data Imputation. International Journal of Engineering Research and Development (or ISO Abbreviation – Int. J. Eng. Res. Dev.), 2012, 5, 1, 5-7.

Other example - page 14, line 510, number 50 in the bibliographic references list – Willett, W. Nutritional Epidemiology. Volume 40, Third Edition. Oxford University Press, 2013, ISBN 978-0-19-975403-8.

The presentation of authors in alphabetical order helps reduce the risk of duplicates substantially and spot missing identifiers easily (when is applicable).

The work also benefits from adequate iconographic support, materialized by five figures. The data included in the figures accurately reflect the main objectives and the results obtained. More than that, the primary material is accompanied by an additional material represented by five figures, which harmoniously complete the database provided by the article.

The authors should pay more attention to the use of certain abbreviations to avoid confusion; basically, all abbreviations are to be used in the text-only after at least one mention made in extenso.

The obtained results are interpreted correctly, and their practical value is visible.

The graphical representation of the results is adequate. As for the paper's grammar, the text is very well written. Consequently, I have only a few recommendations, as follows:

Page 1, line 25 – replace "important achievement" with "important achievements";

Page 6, line 241 – replace "Several information" with "Some information."

Minor corrections and clarifications notwithstanding, the authors’ work and obtained results are highly commendable. They add significant value to the paper and may constitute a launching pad for further valuable studies.

The article can be accepted and published in the Nutrients journal if the authors verify the paper and perform the required corrections.

Best Regards,

Reviewer

Author Response

Dear Editor,

Please consider the revised version of the manuscript entitled “The Application of Clustering on Principal Components for Nutritional Epidemiology: a Workflow to Derive Dietary Patterns” in which we have considered all comments and suggestions from reviewers. This letter is intended for the convenience of the editor and reviewers and contains the list of the requested changes. The following list of changes and answers to comments of Reviewers addresses all revisions made in the manuscript (in red font).

Reviewer 3

R: Esteemed Authors,

It has been a great honor, as well as a pleasantly challenging activity, to review the article entitled “The Application of Clustering on Principal Components for Nutritional Epidemiology: a Workflow to Derive Dietary Patterns.”

Agricultural and food production has significantly changed in the last 20 years. Obviously and logically, increasing population and consumption put pressure on the world's food supply. According to the data of the Food and Agriculture Organisation (FAO), by the effect of demographic growth and changes in diets and incomes, the demand for food will likely grow by 70% until 2050. The current outlook of the increasingly global market is marked by considerable uncertainties of supply linked to unpredictable economic, political, climatic, and biological developments. This implies a need for accelerated agricultural production growth in developing countries. The list of new challenges is open and includes the most unexpected situations, from new crop and animal diseases to significant climate change and emerging diseases. Current agriculture consumes enormous resources for development: over 70% of freshwater reserves are used for agriculture. At the same time, agriculture represents an essential threat to the environment: it is responsible for polluting almost 80% of the oceans and freshwater reserves. On the other hand, products of animal origin are among the top products in terms of greenhouse gas emissions, and approximately one-third of the food produced in the world for human consumption — nearly 1.3 billion tons — gets lost or wasted every year. With a few exceptions, food production is growing. However, the challenges to primary production are increasing. Concerning organic farming in the European Union, the demand for such products is constantly growing, and the implementation mechanisms need to be continuously adjusted. The recent reforms of CAP (Common Agricultural Policy) and other EU policies and international and bilateral trade negotiations take into account the objective of global food security. The Joint Research Center (JRC) of the European Commission is involved in the impact assessment of policies regarding food security. Also, the potential trade agreements through economic modeling and the global CGE (Computable General Equilibrium) models assess the economy-wide impacts of the trade policy changes. All these changes are affecting all sectors of the partners. Besides, the global partial equilibrium models simulate the consequences incurred by the agricultural areas of the partners. The situation is much more complicated concerning organic products than conventional products. Given that organic products are among the top products subject to various frauds, consumer confidence appears as an essential element of the development of the food chain. Theory-wise, the paper will likely elicit specialists' interest in consumer behavior, sociology, sustainable development of agriculture, nutrition, food processing, food consumption, public policies, and public health.

Answer: We are very grateful for this interesting and motivating introduction on nutrition, sustainable development of agriculture and food processing, and consumer behavior, which contextualizes our work. We would like to take this opportunity to thank the Reviewer for his/her comments and suggestions which helped us in improving our manuscript.

R: The paper is well structured and possesses an appreciable novelty character. The main components of the article – Introduction, Materials and Methods, Results, Discussion, and Conclusions - are organized judiciously and directly linked to one another. The documentation is adequate, and the provided scientific results are precise. The goal of the conducted research is well-specified and delineated. The working protocol is appropriate, and the analysis methods are coherent with the proposed objectives. The bibliography of the paper is generous. What is even more relevant for the overall quality of the article, despite the appreciable number of bibliographic sources, all the authors in the bibliographic reference list are quoted in the text of the material (without exception). The article is very well documented, and most bibliographic references are recent and very recent.

A: We thank again the Reviewer for his/her positive comments on our work.

R: I would advise the authors to be more careful concerning the bibliography: it is preferred to mention the authors in alphabetical order, and references without specified authors are cited at the end of the list of references in chronological order. I also recommend using a single system not only in citations but also when it comes to journals. I am referring here mainly to mentioning the following elements for each article consulted: journal, volume, issue, and pages. Supplementary, the DOI may also be noted if the authors desire, but the essential descriptive elements are the previously mentioned ones. Also, to avoid confusion, it is recommended to accurately mention the article's descriptive elements - for example - the additional mention of the article number, where the situation requires it.

For example – page 14, line 506, number 47 in the bibliographic references list – Malarvizhi, R., Thanamani, A.S. K-Nearest Neighbor in Missing Data Imputation. International Journal of Engineering Research and Development (or ISO Abbreviation – Int. J. Eng. Res. Dev.), 2012, 5, 1, 5-7.

Other example - page 14, line 510, number 50 in the bibliographic references list – Willett, W. Nutritional Epidemiology. Volume 40, Third Edition. Oxford University Press, 2013, ISBN 978-0-19-975403-8. The presentation of authors in alphabetical order helps reduce the risk of duplicates substantially and spot missing identifiers easily (when is applicable).

A: Dear Reviewer, we agree with your comment, however, we have formatted the references according to the Journal style using Endnote as reference management software. Please consider that we are required to follow the Journal instructions. Anyway, we have revised some of the references where possible. 

R: The work also benefits from adequate iconographic support, materialized by five figures. The data included in the figures accurately reflect the main objectives and the results obtained. More than that, the primary material is accompanied by an additional material represented by five figures, which harmoniously complete the database provided by the article.

A: We thank again the Reviewer for his/her positive comments on our work.

R: The authors should pay more attention to the use of certain abbreviations to avoid confusion; basically, all abbreviations are to be used in the text-only after at least one mention made in extenso.

A: As suggested, we have revised the abbreviations throughout the text.

R: The obtained results are interpreted correctly, and their practical value is visible. The graphical representation of the results is adequate. As for the paper's grammar, the text is very well written. Consequently, I have only a few recommendations, as follows:

A: We thank again the Reviewer for his/her positive comments on our work.

R: Page 1, line 25 – replace "important achievement" with "important achievements”.

Page 6, line 241 – replace "Several information" with "Some information."

A: As suggested, we have revised these points in the manuscript.

R: Minor corrections and clarifications notwithstanding, the authors’ work and obtained results are highly commendable. They add significant value to the paper and may constitute a launching pad for further valuable studies. The article can be accepted and published in the Nutrients journal if the authors verify the paper and perform the required corrections.

A: We thank again the Reviewer for his/her positive comments on our work.

Round 2

Reviewer 1 Report

The paper is successfully improved.

Reviewer 2 Report

Thanks a lot for the reply. I appreciate the novel thinking and nice attempt from your side. However, it is quite normal to choose different methods at different steps, according to the raw data quality. If this paper focused on a real study / use case, and the method (combinatioon) was chosen according the corresponding data features (data quality, data size, etc.), it may sound more appropriate and reasonable.

However, as a methodology paper, more theoritical discussion (equations could be helpful to explain the theories), or real use cases would be necessary to test and evaluate the method, it would also be needed when other researchers reuse your method for there cases.

I would consider this study a commentary, or a perspective (short discussion), rather than a real research paper. 

As a result, I would not recommend to publish this paper in the current version.